# Principal Component Analysis of Alternative Splicing Profiles Revealed by Long-Read ONT Sequencing in Human Liver Tissue and Hepatocyte-Derived HepG2 and Huh7 Cell Lines

**DOI:** 10.3390/ijms242115502

**Published:** 2023-10-24

**Authors:** Elizaveta Sarygina, Anna Kozlova, Kseniia Deinichenko, Sergey Radko, Konstantin Ptitsyn, Svetlana Khmeleva, Leonid K. Kurbatov, Pavel Spirin, Vladimir S. Prassolov, Ekaterina Ilgisonis, Andrey Lisitsa, Elena Ponomarenko

**Affiliations:** 1Institute of Biomedical Chemistry, Pogodinskaya Street 10, 119121 Moscow, Russia; lizalesa@gmail.com (E.S.); ministreliya13113@gmail.com (A.K.); radkos@yandex.ru (S.R.);; 2Department of Cancer Cell Biology, Engelhardt Institute of Molecular Biology, Russian Academy of Sciences, Vavilova 32, 119991 Moscow, Russia

**Keywords:** nanopore sequencing, alternative splicing, human liver tissue, Huh7 and HepG2 cell lines, transcriptome, pharmacogenes

## Abstract

The long-read RNA sequencing developed by Oxford Nanopore Technology provides a direct quantification of transcript isoforms. That makes the number of transcript isoforms per gene an intrinsically suitable metric for alternative splicing (AS) profiling in the application to this particular type of RNA sequencing. By using this simple metric and recruiting principal component analysis (PCA) as a tool to visualize the high-dimensional transcriptomic data, we were able to group biospecimens of normal human liver tissue and hepatocyte-derived malignant HepG2 and Huh7 cells into clear clusters in a 2D space. For the transcriptome-wide analysis, the clustering was observed regardless whether all genes were included in analysis or only those expressed in all biospecimens tested. However, in the application to a particular set of genes known as pharmacogenes, which are involved in drug metabolism, the clustering worsened dramatically in the latter case. Based on PCA data, the subsets of genes most contributing to biospecimens’ grouping into clusters were selected and subjected to gene ontology analysis that allowed us to determine the top 20 biological processes among which translation and processes related to its regulation dominate. The suggested metrics can be a useful addition to the existing metrics for describing AS profiles, especially in application to transcriptome studies with long-read sequencing.

## 1. Introduction

Principal component analysis (PCA) is one of the oldest and most widely used mathematical tools to reduce the dimensionality of large datasets, allowing us to increase the data interpretability but, at the same time, minimize information loss [1,2]. In application to omics technologies, PCA allows one to visualize high-dimensional data, generated by these technologies, by projecting them into a 2D or 3D space [3]. The visualization is crucial for detecting intrinsic structure of the data, such as clusters and outliers. In transcriptome-wide studies, PCA was recruited to dissect both gene expression [4] and alternative splicing [5] profiles.

Alternative splicing (AS) is a highly specialized processing of pre-mRNAs which allows organisms to enhance transcriptome and proteome diversity [6,7]. Over 90% of transcripts may undergo alternative RNA processing by the skipping/inclusion of exons or retention of introns via differential selection of splice sites in pre-mRNAs, enriching the phenotypic diversity [7,8]. AS dysregulation plays an important role in pathogenesis [9]. Aberrations in AS have been linked, e.g., to various oncogenic processes including tumor progression, metastasis, and therapy resistance [10].

The advent of the high-throughput sequencing of short cDNA fragments (RNA-seq) greatly boosted the transcriptome-wide analysis of AS [6,11]. However, despite successfully detecting AS events, RNA-seq is still limited due to a reliance on short reads since, during data analysis, the transcript assembly software faces difficulties in determining the connectivity between exons when one gene has many isoforms and the assembling of short reads into transcript isoforms becomes prone to errors [6]. As a way around this, metrics were suggested to quantify AS without assembling short reads into transcript isoforms, including ‘exone usage’, ‘Percent Spliced-In’ (PSI), and ‘Percent of Intron Retention’ indexes [12,13,14]. Among them, the PSI index was employed in PCA to dissect AS profiles, using the short-read RNA-seq data [15].

The emergence of long-read RNA sequencing such as that developed by the Oxford Nanopore Technologies (ONT), with long reads spanning multiple exons, opened new opportunities to study AS with less ambiguity, by directly identifying and quantifying transcript isoforms [7]. In addition, ONT sequencing can be carried out in the format of direct RNA sequencing, thus avoiding potential biases which can result from the amplification of cDNA fragments (though at the expense of sequencing yield). This allows one to reveal splicing patterns in terms of the relative abundance of transcript isoforms (expressed using, e.g., the ‘transcripts per million’ metric, TPM) and/or as the number of transcript isoforms per each and every gene.

In this study, we aimed to evaluate whether such a metric as the number of transcript isoforms per gene, which will be hereinafter referred to as ‘degree of splicing’ (DS), can be utilized to visualize differences in AS profiles between cell and tissue types by means of PCA. Samples of human liver tissue and hepatocyte-derived HepG2 and Huh7 cultured cells were employed for this purpose and the extracted mRNA was subjected to long-read ONT sequencing. HepG2 and Huh7 are hepatocyte-derived cell lines originated from human hepatoblastoma (HBL) and hepatocellular carcinoma (HCC) tumors, respectively [16,17], and commonly used for studying processes associated with the malignant transformation of hepatocytes (e.g., [17,18,19]). Since liver cells are responsible for the metabolism of the most drugs, alongside analyzing the whole transcriptome, we also applied PCA, using DS as a metric, to a particular group of genes known as ‘pharmacogenes’ [20]. Pharmacogenes are involved in the processing of drugs and are composed, among others, of genes encoding Phase I (e.g., cytochrome P450) and Phase II (e.g., UDP-glucuronosyltransferase) enzymes.

## 2. Results

### 2.1. The Transcript Isoforms Abundance Profiles in Liver Tissue and Hepatocyte-Derived Cultured Cells

The number of transcript isoforms detected for a given gene depends on the sequencing depth—the parameter hardly controlled in the ONT sequencing. The sequencing depth greatly varied among the analyzed biospecimens (Appendix A). To alleviate the impact of variation in sequencing depth on DS, we systematically varied the TPM threshold values to cut off the “noisy” low-abundance transcript isoforms. Prior to setting TPM threshold values, we as a first step compared the distributions of the number of transcript isoforms by their abundance for the studied types of biospecimens, using all detected transcript isoforms. The entire range of transcripts’ abundance determined by the ONT sequencing was split into uneven bins whose means differed by a factor of about three to visually enlarge the range of low- and medium-abundance transcripts. The range started with the TPM value of 0.1 since for all biospecimens used in the analysis there were no transcript isoforms found with an abundance below this value. To construct distributions, we grouped transcript isoforms into ‘bins’ representing their abundance (in TPM) for each biospecimen and then averaged over all biospecimens of a given type for each bin. The result is presented as a histogram in Figure 1. As seen, the distributions are similar in general. Yet, the average abundance of moderately-expressed transcript isoforms (10 ≤ TPM ≤ 100) is statistically lower in the liver tissue compared to that in HepG2 and Huh7 cells. Also, there is a considerable variability in the abundance of low-expressed transcripts (TPM ≤ 1) between biospecimens for HepG2 cells and liver tissue as manifested by the large confidence intervals (Figure 1). Moreover, for the liver tissue biospecimens, there is no transcripts’ isoform whose abundance fell into the bin (0.1, 0.316]. This is most probably a consequence of the lower sequencing outputs for the biospecimens of liver tissue compared to those we obtained for cultured cells in our sequencing experiments (Appendix A). Interestingly, if transcript isoforms abundances are averaged in another way, viz. by first calculating means for each transcript isoform over biospecimens of a given type and then plotting the histogram by using these mean values of isoforms’ abundance (Appendix A), the number of isoforms falling into the bin (0.1, 0.316] increases dramatically compared to that in Figure 1. This is likely due to a great variability in the abundance of low-expressed transcript isoforms between biospecimens of the same type so that, depending on the way of averaging, they fall into different TPM bins.

Second, we considered how the cutting off of low-abundance transcript isoforms by increasing the TPM threshold can affect the overall splicing (defined here as the number of all transcript isoforms with an abundance above the TPM threshold, averaged over biospecimens of a given type, divided by the number of genes expressing them). We have extended this analysis up to a threshold value of 100. The results for the TPM thresholds of 0.1, 1, 10, and 100 are presented in Appendix A. It turned out that the overall splicing is quite different for liver tissue, HepG2, and Huh7 cells when the TPM threshold equals 0.1 (1.6, 2.1, and 1.9, respectively) but becomes indistinguishable between the studied types of biospecimens at the TPM thresholds of 10 and 100 (1.3 and 1.2, respectively). However, in the last case, the quantity of transcript isoforms left for analysis is dramatically reduced (by about 20-fold compared to TPM > 0.1; Appendix A). Thus, three TPM thresholds, 0.1, 1, and 10, were used in the further analysis. For the TPM threshold of 10, when the overall splicing is similar for all types of biospecimens, the AS profiles are expected not to be influenced by the sequencing depth.

### 2.2. Principal Component Analysis of Alternative Splicing for Whole Transcriptome and Subset of Pharmacogenes in Liver Tissue and Hepatocyte-Derived Cultured Cells

We applied PCA to the datasets where each gene was characterized by its DS value as follows: the DS of a gene was set to equal the number of isoforms expressed with the abundance above the TPM threshold for each of the biospecimens; if there were no transcript isoforms with an abundance above the TPM threshold for that gene in a particular biospecimen, then the gene’s DS value in that biospecimen was set to equal zero (Appendix A). The result for the TPM threshold of 0.1 (when, in fact, all detected transcript isoforms expressed by a gene are counted) is presented in Figure 2a. As seen, the biospecimens are well clustered by PCA according to their type. When the TPM threshold was increased to exclude the potentially most variable transcript isoforms from the analysis, the clear clustering of biospecimens according to their type was still preserved (Figure 2b,c). The clustering also remained obvious for TPM > 10 where the overall splicing became indistinguishable between the studied types of biospecimens (Appendix A).

As the next step, we conducted PCA under more stringent conditions by excluding from the analysis all genes which had a DS value equal to zero for, at least, one of the biospecimens. The motivation was to evaluate whether splicing profiles for subsets of genes which are expressed in all biospecimens under study are still different so as to allow for discriminating biospecimens between their types (liver tissue, HepG2, and Huh7 cells). Figure 2d–f shows that the clustering of biospecimens according to their type remains obvious for all TPM thresholds tested. It is worth noting that, in this case, the number of genes included in the analysis is considerably reduced—to 6666, 6425, and 2599 for TPM thresholds of 0.1, 1, and 10, respectively.

Among 384 genes listed as pharmacogenes in [20], the expression of 323 was detected in at least one of the biospecimens in our sequencing experiments (Appendix A). We constructed datasets composed of these pharmacogenes characterized by DS values in each biospecimen for TPM thresholds of 0.1, 1, and 10 (Appendix A). The results of PCA of these datasets are shown in Figure 3a–c. As seen, the biospecimens are clustered according to their type at all TPM thresholds tested. However, when pharmacogenes are excluded from the dataset if their DS values equal zero even in one biospecimen, the clustering worsens dramatically (Figure 3d–f. It should be noted that merely 115 (TPM > 0.1), 111 (TPM > 1), and 56 (TPM > 10) pharmacogenes are included in the analysis in this case. Nonetheless, there is a separation between biospecimen clusters of Huh7 cells and liver tissue at the TPM threshold of 0.1 along PC2 (Figure 3d) and between clusters for liver tissue and hepatocyte-derived cells at the TPM threshold of 10 on PC1 (Figure 3f). Interesting, all three types of biospecimens show separated clusters along PC1 (Figure 3e).

Since it is known that gene expression can be considered as an integrated dynamical system and that any random choice of genes gives more or less the same results (at least for gene expression in terms of transcript abundance [21,22]), we applied PCA to subsets of genes randomly selected from the gene sets used for PCA in Figure 2c,f. The sizes of these random subsets match those of the pharmacogene subsets used for PCA in Figure 3c,f, respectively. The result is presented in Appendix A. As in the case of gene expression in terms of transcripts’ abundance, the AS profiles in terms of DS values have also demonstrated the ability to group biospecimens according to their type in PCA for randomly chosen subsets of genes. This is in contrast to pharmacogenes, for which AS profiles did not allow us to clearly group biospecimens in the case when only genes with no zero DS values are involved in the analysis (only moderately-expressed splice variants are taken into account) (Figure 3f vs. Appendix A).

### 2.3. Subsets of Genes Determining Differences in Alternative Splicing Profiles for Liver Tissue and Hepatocyte-Derived Cultured Cells

To determine subsets of genes whose differences in AS contribute the most to the biospecimens’ distribution to clusters in PCA, we retrieved loadings for PC1 and PC2 from the PCA data. The PCA data used correspond to the case represented by Figure 2f when no genes with zero DS values in at least one biospecimen are included in the analysis (the more constrained version of the suggested metric) and the TPM threshold is set as 10 (the AS profiles are expected not to be influenced by the sequencing depth). The loadings can be understood as the weights for each original variable when calculating the principal component [1,2]. Loadings vary in the range of −1 to 1, and the further the loading value is from zero, the more strongly the variable (ENGS identifier in our case) influences the component.

The values of loadings are presented in Appendix A and were used to construct box plots for PC1 and PC2 (Appendix A). Genes (ENGS identifiers) which constitute the outliers (data points located outside the whiskers of the box plot) were chosen for a further consideration and provided in Appendix A (400 and 655 genes for PC1 and PC2, respectively). Appendix A contains ENGS identifiers as rows and biospecimens as columns with the number of transcript isoforms (DS values) for each of the listed gene in a given biospecimen.

The defined sets of genes contributing the most to the biospecimen clustering (Figure 2f) were used to look at biological pathways which are influenced by differences in AS patterns of these genes in human liver tissue and hepatocyte-derived malignant cells. We applied the one-way ANOVA test to those sets of genes and selected only genes with the mean DS values (obtained by averaging DS values over all biospecimens of a given type) which statistically significantly differ between types of biospecimens (*p*-value < 0.01). Finally, we arrived at two subsets of genes (163 and 174 ENGS identifiers for PC1 and PC2, respectively; Appendix A) which were subjected to gene ontology (GO) analysis. The results are presented in Figure 4 as the top 20 biological pathways revealed by GO analysis. As seen, for both components, translation appeared as the top biological pathway influenced by the statistically significant differences in AS profiles, responsible for the distribution of biospecimens to clusters according to their type.

### 2.4. Transcript Isoforms Involved in Translation and Characteristic for Liver Tissue, HepG2 or Huh7 Cells

To see which transcript isoforms involved in translation-related pathways can be considered as ‘characteristic’ (detected only in one type of biospecimen at a given TPM threshold), we selected genes from Appendix A participating in the transcription pathway (Figure 4). Additionally, we included genes from Appendix A involved in other biological pathways related to the regulation of translation such as “Translation factors”, “Mitochondrial translation initiation”, and “DGCR8 multiprotein complex” (Figure 4). In overall, 31 genes (109 transcript isoforms) from Appendix A were selected as involved in translation related pathways and are presented in Appendix A.

Appendix A shows a heatmap of the distribution of the number transcript isoforms by biospecimens for each of 31 selected genes. There was a striking difference between the diversity of transcript isoforms in liver tissue and hepatocyte-derived malignant cells: the number of transcript isoforms in liver tissue is systematically lower. For HepG2 and Huh7, the diversity was rather similar; however, the genes RPL23A and DDX5 (coding a ribosomal protein and RNA helicase, respectively) exhibited an especially high level of splicing (6 transcript isoforms) in Huh7 cells. The heatmap in Appendix A presents 109 transcript isoforms detected in liver tissue, HepG2, and Huh7 cells at the TPM threshold of 10: there were 18 transcript isoforms observed only in one type of biospecimen studied (characteristic isoforms), which are listed in Table 1. Among these transcript isoforms, 6 isoforms code for protein isoforms which were not detected to date (“computationally predicted”), 6 code for canonical protein isoforms (“canonical forms”), and the other 6 protein isoforms are non-canonical but were experimentally detected as proteins (“reviewed protein isoforms”). Thus, 6 out of 18 characteristic transcript isoforms can produce different protein products involved in translation-related processes and contributing to phenotypes of normal and malignant hepatocytes.

## 3. Discussion

In the short-read RNA-seq data analysis, it is commonly accepted to describe AS profiles by using such metrics as ‘exone usage’ or PSI index [11,12,13,14]. In fact, they are a substitute of truly AS profiling in terms of splice variants (transcript isoforms) per gene. These metrics continued to be used in the ONT data analysis (e.g., [23,24,25]) despite the fact that, in long-read ONT sequencing, the standard treatment of raw sequencing data directly provides the abundance of each and every transcript isoform and, consequently, allows us to easily calculate the number of splice variant (transcript isoforms) expressed by each gene. That makes the number of transcript isoforms per gene an intrinsically suitable metric for alternative splicing (AS) profiling in the application to a particular type of RNA sequencing such as ONT sequencing.

Clearly, the suggested metric—the number of transcript isoforms per gene, or, in short, the degree of splicing—has to allow, as a minimum, for distinguishing between various cells and/or tissue types, based on differences in their AS profiles. To test whether the suggested metric allows us to discriminate cells/tissues of various types, we applied PCA to AS profiles revealed by ONT sequencing and expressed in terms of DS. We employed biospecimens of three types—from normal human liver tissue and from two hepatocyte-derived malignant cell lines. Since the dataset for PCA was composed of the RNA sequencing results for a rather limited number of biospecimens (11 samples in total), for such testing the types of biospecimens should be expected to have quite different AS profiles. Indeed, malignant transformation is known to be associated with alterations in AS—tumors have, on average, up to 30% more AS events than normal tissues [26,27]. The HepG2 and Huh7 cell lines employed in our study as examples of hepatocyte-derived cells are originated from the malignant human liver tumors of different types, HBL and HCC, respectively. While HBL results from the malignant transformation of pluripotent hepatic stem cells, HCC results from that of differentiated hepatocytes [28]. Thus, one may expect that the types of biospecimens used in our study will be characterized by quite different AS patterns.

Indeed, using AS profiles described in terms of DS, PCA clearly grouped all biospecimens tested into three well-defined and separated clusters, both for the transcriptome-wide analysis (Figure 2a) and for the subset of pharmacogenes (Figure 3a). However, the detection of low-abundance transcript isoforms inevitably depends on the transcriptome sequencing depth (the overall number of mapped reads) [29] which, in the case of direct RNA ONT sequencing, can substantially vary and is hardly controlled (since it is determined to a large extent by the initial quality of the sequencer’s flow cell as supplied by the manufacturer). Thus, variations in sequencing depth between biospecimens (Appendix A) can influence the revealed AS patterns, making it hard to differentiate the biological variability in AS profiles from that caused by technical reasons. To overcome this problem, we applied PCA to datasets obtained by systematically varying a TPM threshold. To set a TPM threshold is a routine practice in RNA sequencing to filter off noisy reads. In our case, such an approach allowed us to cut off the low abundance transcript isoforms detected only at a substantial sequencing depth and, therefore, to minimize the variability in AS patterns, introduced due to technical reasons. Upon the increase in the TPM threshold to 10, the overall splicing becomes similar for the biospecimen types (Appendix A). One cannot rule out that datasets obtained by elevating the TPM threshold can also lose partially the biologically relevant information on AS. Nonetheless, the remaining differences between AS patterns, which can now be considered as biologically relevant, allow for a clear grouping of biospecimens into clusters according to their types (Figure 2c).

By setting the DS of a gene equal to zero when no transcript isoforms are detected or expressed above a TPM threshold, we assign a zero value for a biospecimen on a particular axis in the multidimensional space of ‘gene names’, which is composed of all genes showing expression in at least one biospecimen. However, rigorously, splicing is a characteristic of an expressing gene while the zero value assumes that a gene does not express any transcript isoforms. Formally, we can describe AS patterns by including all genes, regardless of whether they are expressing transcripts in a detectable amount or not (in the latter case, their splicing is taken as zero). Yet, it seemed interesting to us to examine how the biospecimens would be grouped by PCA if only genes whose expression is detected in all biospecimens (with no zero DS values in a dataset) are taken into account. We found that the clustering of biospecimens is still observed (Figure 2d–f), thus indicating that the more constrained version of the suggested metric allows one to distinguish the characteristic patterns of AS for HepG2 and Huh7 cell lines and liver tissue.

In the case of the subset of pharmacogenes, the clear clustering of biospecimens was also observed (Figure 3a–c). The Hep G2 cell line was considered for a long time as a suitable in vitro model for studying human drug metabolism but was later subjected to criticism due to the different spectrum and low expression of cytochrome P450 compared to normal hepatocytes [16,17,30]. As our results demonstrate, the AS patterns of pharmacogenes’ expression, defined in terms of DS, do differ in normal liver tissue and Hep G2. This also holds for Huh7 cells. Moreover, AS patterns appear to vary considerably between HepG2 and Huh7 cells as well, likely reflecting the different origins of these cell lines. As mentioned in Section 2.2, any randomly chosen subset of genes can demonstrate a behavior similar to that of the original set, at least in terms of transcript abundance [21,22]. However, if the more constrained metric is used (with no DS values of zero allowed), the clustering becomes strongly dependent on the TPM threshold and worsens in general (Figure 3d–f). At the same time, the randomly chosen subset of the matching size grouped biospecimens into clear clusters (Appendix A). In fact, it may reflect that, among the pharmacogenes steadily expressed (detected in all biospecimens) in liver tissue, HepG2, and Huh7 cells, there are no groups of genes whose AS patterns could allow one to clearly discriminate all three biospecimen types by projecting the multidimensional datasets into a 2D space. This appears quite possible since the overall explained variance for PC1 and PC2 is below 60% at all TPM thresholds (Figure 3d—f).

Investigating differences in the profiles of splice forms between the hepatic cell lines HepG2 and Huh7 and normal liver tissue, one cannot ignore the fact that the cell lines are of cancerous origin. AS is known to be one of the key drivers of the formation of a cancer phenotype through the production of aberrant splice forms, the formation of splice forms with an altered translation rate, or direct influence on the diversity of proteoforms [27,31]. From Appendix A, about 41% and 26% of the genes for which AS was associated with the clustering of biospecimens of liver tissue, HepG2, and Huh7 cells in the first and second components, respectively, statistically significantly differed in the number of splice variants (transcript isoforms). The results of our analysis demonstrate that AS in the studied types of biospecimens differed significantly in genes involved in the mRNA translation process (Figure 4, pathways R-HSA-72766 “Translation”, WP107 “Translation factors”, R-HSA-5368286 “Mitochondrial translation initiation”). Numerous studies have shown that the process of mRNA translation in cancer cells differs significantly from that in normal cells [32,33]. However, the focus of these studies was commonly on the formation of defective splice forms. As our results show, the possible cause of such differences may also be a different profile of splice forms of genes involved in translation. In addition, it has been shown that the rRNA/protein composition of ribosomes may differ depending on the cell type and AS may also contribute to the formation of tissue-specific ribosomal proteins [34,35]. GO analysis of genes statistically significantly differing in the number of splice forms revealed that genes of the DGCR8 protein complex involved in the maturation of microRNAs and the formation of spliceosomes also differed in the number of detected splice forms, which can affect both the process of AS and mRNA translation [36,37].

To date, considerable efforts have been put into finding novel transcript isoforms that can potentially contribute to cell identity (cell phenotype) (e.g., [6] and references therein). However, the cell phenotype can also be influenced by differences in AS profiles via a variation in composition of known transcript isoforms. As our results show, the set of genes revealed by the extended PCA as those whose differential splicing appears to considerably contribute to the phenotypes of the studied biospecimens and by GO analysis as involved in translation-related processes contains genes expressing transcript isoforms observed (at a given TPM threshold) only in one type of biospecimen studied (which we here referred to as characteristic isoforms). Some of these isoforms code for predicted protein isoforms not actually detected to date (Table 1). Yet, two thirds of them (12 transcript isoforms, Table 1) code for either canonical or non-canonical protein isoforms which are expressed and that can potentially influence the particular biological pathway and, consequently, contribute to the observed phenotypes of the studied biospecimens.

To sum up, in the case of long-read ONT sequencing, the mapping of reads to a reference genome/transcriptome allows for the direct quantification of transcript isoforms of a given gene. Thus, in the application to this particular type of RNA sequencing, the degree of splicing defined here as the number of transcript isoforms per gene appears as an intrinsically suitable metric for AS profiling. By using this simple metric and recruiting PCA as a tool to visualize the high-dimensional transcriptomic data, we were able to group biospecimens of normal human liver tissue, HepG2, and Huh7 cells, which are expected to differ in their patterns of AS, into clear clusters in a 2D space. Furthermore, based on PCA data, we selected subsets of genes most contributing to biospecimens’ grouping into clusters according to their type. The further analysis of these subsets allowed us to determine that those genes are involved into the mRNA translation process and its regulation. The suggested metrics can be a useful addition to the existing metrics for describing AS profiles, especially in application to transcriptome studies with the long-read sequencing.

## 4. Materials and Methods

### 4.1. Tissue Collection and Cell Culturing

Biospecimens of human liver were collected at autopsy from 38, 54, and 65 years old male donors with informed consent from the donor’s representatives. The tissue collection was approved by the N.I. Pirogov Russian State Medical University Ethical Committee (Protocol #3; 15 March 2018). The donors were HIV- and hepatitis-free, and the sections had no histological signs of liver diseases. The postmortem resected samples were immediately placed into the RNAlater RNA Stabilization Solution (Thermo Fisher Scientific, Waltham, MA, USA) and stored at −20 °C until further use. 

The HepG2 and Huh-7 cells were purchased from Merck (Darmstadt, Germany) and Thermo Fisher Scientific, respectively. Both cell lines were cultivated under identical conditions to about 80% confluence. The culturing conditions are provided in [38]. Cells were harvested, washed with phosphate-buffered saline (Merck), and used for RNA isolation.

### 4.2. RNA Isolation, Sequencing Library Preparation, and Long-Read ONT Sequencing

Biospecimens of normal human liver tissue from three donors (one sample of each), HepG2 (5 samples), and Huh7 (3 samples) cultured cells were used to isolate total RNA with an RNeasy Mini Kit (Qiagen, Hilden, Germany) according to the manufacturer’s manual. RNA quality was assessed using a Bioanalyzer 2100 System (Agilent Technologies, Palo Alto, CA, USA). The RNA integrity numbers were 7.8 or higher for all preparations of total RNA. The mRNA extraction was carried out with a Dynabeads™ mRNA Purification Kit (Thermo Fisher Scientific, Waltham, MA, USA), following the manufacturer’s instructions. mRNA was quantified using a Qubit 4 fluorometer and a Qubit RNA HS Assay Kit (Thermo Fisher Scientific). The mRNA preparations were either used immediately for the preparation of sequencing libraries or frozen and stored at −80 °C until further use.

To prepare sequencing libraries, a Direct RNA sequencing kit (SQK-RNA002, ONT, Oxford, UK) was used strictly following the manufacturer’s protocol. The long-read sequencing was carried out on a MinION nanopore sequencer (ONT) in 48 h single runs, using FLO-MIN106 flow cells. The row data were processed using the guppy_basecaller 3.1.5 software (ONT) as described in [39]. During processing, the data were filtered by the guppy_basecaller software with a quality score parameter >7.0. The quality control of reads was performed with the MinIONqc.R script. Mapping was carried out with the long-read aligner minimap2 v.2.17 [40] in ‘-ax splice junc-bed’ mode, using the Gencode38 genome assembly (release GRCh40). The number of mapped reads for each biospecimen is presented in Appendix A. Transcript abundance was quantified in TPM with the Salmon 0.12/1.1.0 software [41]. The sequencing data are deposited to the NCBI Sequence Read Archive (PRJNA765908, PRJNA893571, PRJNA635536).

### 4.3. Principal Component Analysis

To prepare datasets for PCA, the Salmon Quant output files with the evaluated genes’ expression were assembled into an Excel table where each gene was characterized by a set of detected transcript isoforms with corresponding TPM values. The table contained 13 columns (columns of ENSG and ENST identifiers, as well as 11 columns with TPM values of transcript isoforms for each biospecimen) and 87,814 rows with different ENST identifiers. The ENST identifiers corresponding to transcripts of protein-coding genes were extracted for further analysis. The table was then converted into another Excel table (Appendix A) where each protein-coding gene (ENSG identifier) was characterized by the number of transcript isoforms assigned to that gene (the degree of splicing) for each biospecimen tested. The table contained sheets (datasets) with data for TPM thresholds of 0.1, 1, and 10. In each dataset, only transcript isoforms with an abundance above the particular TPM threshold were counted. If no transcript isoforms were detected for a gene at all or if their abundance was below the TPM threshold, then the zero value of degree of splicing was assigned to that gene. The obtained datasets were used as input for transcriptome-wide PCA. The datasets for pharmacogenes listed in [20] were created by extracting rows with the corresponding ENSG identifiers from the transcriptome-wide datasets and used in PCA.

PCA was conducted with a script written using options of ‘scikit-learn’ [42]—a Python module accessed at https://scikit-learn.org (accessed on 10 October 2023). The script is available at https://github.com/lizelx/PCA_alternative_splicing (accessed on 10 October 2023). 

### 4.4. PCA Loadings, ANOVA Test, and Gene Ontology Analysis

The attribute ‘PCA.components_’ of scikit-learn was used to retrieve loadings for Principal Components 1 and 2 from the PCA data at the TPM threshold of 10. The results are presented in Appendix A. To determine genes which are statistically significantly different in the number of transcript isoforms between types of biospecimens, the ANOVA test was conducted using the Python module ‘scipy.stats’ (1.11.3v) and the data from Appendix A for *p*-values of 0.05 and 0.01. The selected genes are listed in Appendix A. The gene ontology (GO) analysis was carried out with the bioinformatics web-tool Metascape, accessed at https://metascape.org/gp/index.html#/main/step1 (accessed on 10 October 2023).

## Figures and Tables

**Figure 1 ijms-24-15502-f001:**
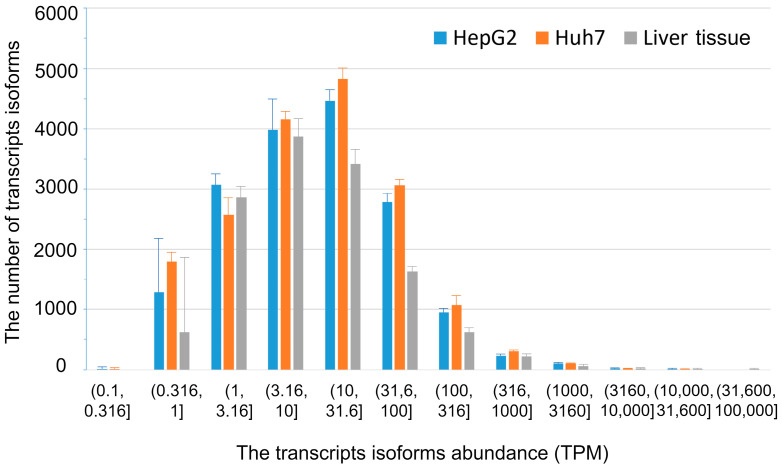
The distributions of the number of transcript isoforms by their abundance. The heights of bars show mean values obtained by averaging the number of transcript isoforms after grouping them into bins over all biospecimens of a given type. The error bars represent 95% confidence intervals. The blue, orange, and grey color correspond to human liver tissue, HepG2, and Huh7 cells, respectively.

**Figure 2 ijms-24-15502-f002:**
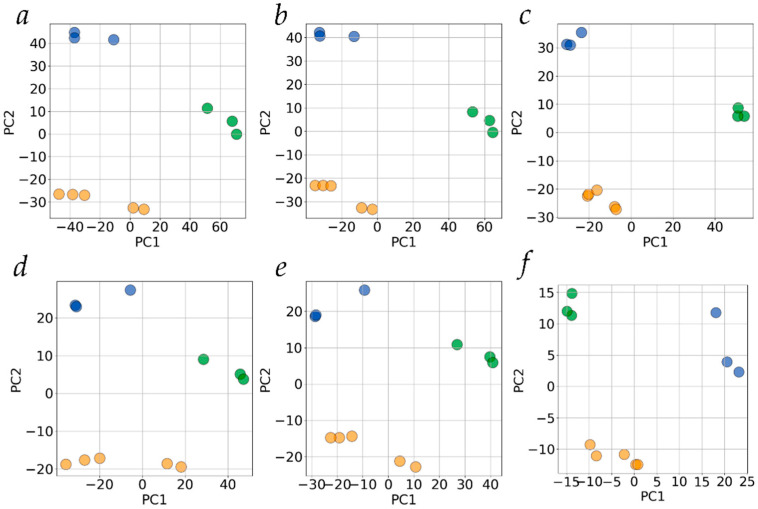
The scores/scores plots. PCA of transcriptome-wide alternative splicing profiles at different TPM thresholds (panels (**a**,**d**)—0.1, (**b**,**e**)—1, (**c**,**f**)—10). Green, yellow, and blue circles represent biospecimens of human liver tissue, HepG2, and Huh7 cells, respectively. Panels a to c show the results of PCA for all genes, panels d to f—for genes which have no zero DS values for any of biospecimens. PC1 and PC2—the first and the second principle components, respectively. The explained variance for PC1 and PC2 was 35.6% and 17.9%, 34.5% and 19.2%, 40.6% and 21.4%, 39.4% and 14.2%, 35.0% and 16.3%, 35.8% and 22.4% for panels (**a**,**b**,**c**,**d**,**e**,**f**), respectively.

**Figure 3 ijms-24-15502-f003:**
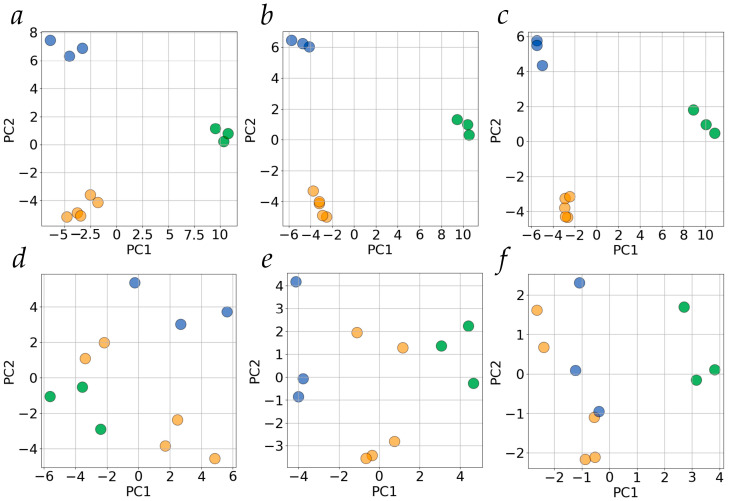
The scores/scores plots. PCA of splicing profiles at different TPM thresholds (panels (**a**,**d**)—0.1, (**b**,**e**)—1, (**c**,**f**)—10) for pharmacogenes. Green, yellow, and blue circles represent biospecimens of human liver tissue, HepG2, and Huh7 cells, respectively. Panels a to c show the results of PCA for all genes, panels d to f—for genes which have no zero DS values for any of the biospecimens. PC1 and PC2—the first and the second principle components, respectively. The explained variance for PC1 and PC2 was 35.1% and 19.9%, 39.3% and 19.5%, 52.9% and 20.0%, 28.9% and 22.5%, 27.3% and 16.8%, 40.6% and 18.8% for panels (**a**,**b**,**c**,**d**,**e**,**f**), respectively.

**Figure 4 ijms-24-15502-f004:**
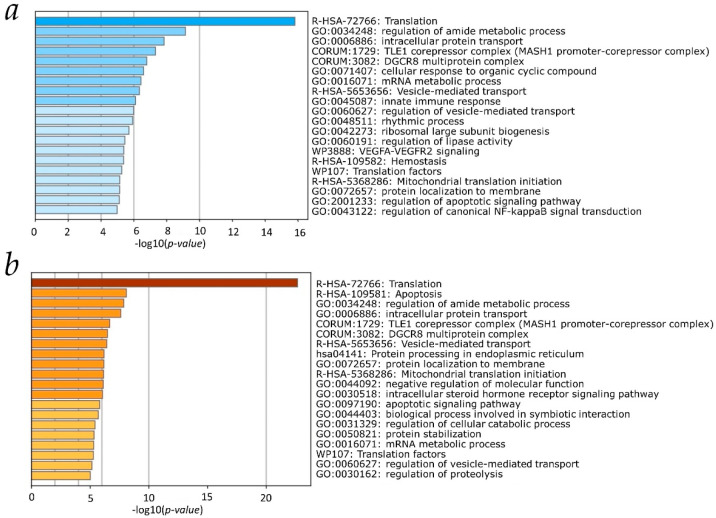
Top 20 biological pathways from GO analysis for sets of genes with statistically significant differences (*p*-value < 0.01) in AS profiles. Panel (**a**,**b**): the results for subset of genes responsible for the observe clustering of biospecimens along the PC1 and PC2 axes, respectively.

**Table 1 ijms-24-15502-t001:** The distribution of characteristic isoforms by biospecimen types and their relation to protein isoforms.

Transcript Isoform	Liver Tissue	Huh7 Cells	HepG2 Cells	UniProt Identifier	Status of Coded Protein
HNRNPR-203 HNRNPR-202 DAP3-214 RPL31-203 EEF1B2-201 RPL32-206 EIF4G1-203 RPL26L1-206 RPS10-207 MRPS33-202 MRPS33-203 MRPL11-203 RPL23A-206 DDX5-223 ILF3-222 EIF3K-213 EIF6-209 EIF6-202	− − − − − − − − − − − − − − − − + −	− − − − + − + − − + + + + + + + − +	+ + + + − + − + + − − − − − − − − −	O43390-2 O43390-1 P51398-2 P62899-2 P24534-1 P62910-1 Q04637-5 Q9UNX3-1 A0A2R8Y7H1 Q9Y291-1 C9JBY7 Q9Y3B7-2 K7EMA7 A0A0G2JLI4 K7EMZ8 K7EK53 P56537-1 P56537-2	Protein isoform Canonical form Protein isoform Protein isoform Canonical form Canonical form Protein isoform Canonical form Predicted Canonical form Predicted Protein isoform Predicted Predicted Predicted Predicted Canonical form Protein isoform

## Data Availability

The data were deposited into the Sequence Read Archive (https://www.ncbi.nlm.nih.gov/sra (accessed on 10 October 2023)) (PRJNA765908, PRJNA893571 and PRJNA635536).

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
