# Peer review of "Principal Component Analysis of Alternative Splicing Profiles Revealed by Long-Read ONT Sequencing in Human Liver Tissue and Hepatocyte-Derived HepG2 and Huh7 Cell Lines"

_ijms, 2023, doi:10.3390/ijms242115502_

Round 1

Reviewer 1 Report

This manuscript entitled "Principal component analysis of alternative splicing profiles revealed by long-read ONT sequencing in human liver tissue and hepatocyte-derived HepG2 and Huh7 cell lines" by Sarygina et al. aimed to determine the alternative splicing profiles in human liver tissues as well as two cell lines by utilizing long-read RNA sequencing technology. Once they got the sequencing data, they analyzed the distribution of spliced mRNAs based on their expression abundance. Then they used PCA to show that samples could be separated by the transcriptome profiles, but could not apply to a specific subset of genes for example, pharmacogenes in this study.

However, the data presented are more like a quality control of the nanopore sequencing technology, which is already known to us. I would expect the authors apply Nanopore sequencing to raise scientific questions and find some new and interesting results to the scientific society, for example, novel splicing isoform that contribute to the cell identity. 

Overall, the conception novelty of the manuscript is low. The data presented are just a QC of nanopore sequencing technology without raising scientific questions and deep investigation of the splicing isoforms got from nanopore sequencing. The current form of the manuscript is hard to be published as a research article. I would suggest the authors take extensive investigation of the splicing isoforms and raise more interesting scientific questions rather than just submit their QC results of the sequencing data. 

Author Response

This manuscript entitled "Principal component analysis of alternative splicing profiles revealed by long-read ONT sequencing in human liver tissue and hepatocyte-derived HepG2 and Huh7 cell lines" by Sarygina et al. aimed to determine the alternative splicing profiles in human liver tissues as well as two cell lines by utilizing long-read RNA sequencing technology. Once they got the sequencing data, they analyzed the distribution of spliced mRNAs based on their expression abundance. Then they used PCA to show that samples could be separated by the transcriptome profiles, but could not apply to a specific subset of genes for example, pharmacogenes in this study.

Indeed, as a first step, we analyzed the distribution of transcript isoforms by their abundance but purely with a purpose of determining TPM threshold we had to set to minimize the impact of sequencing depth on profiles of alternative splicing, revealed by ONT sequencing. We specifically stressed this point upon revision to avoid confusions (lines 84-89 and 134-136 of the revised manuscript). Further, we applied PCA to alternative splicing profiles that means transcriptome profiles expressed in terms of the number of splice variants per a gene but not in terms of the abundance of transcripts isoforms (in TPM). Though we applied PCA to all TPM thresholds tested, in the section “Discussion” we specifically pointed that clustering of biospecimens in PCA is biologically relevant at TPM threshold of 10 (lines 283-288). We would like to specifically mention that we employed a new metric suggested in this manuscript, which we refer to as ‘degree of splicing’. The alternative splicing profiles were described with this metric instead of using such metrics as ‘exon usage’ or PSI index which are rather a substitute for description of alternative splicing. We additionally stressed this issue in the section “Discussion” upon revision (lines 242-251 of the revised manuscript).   

However, the data presented are more like a quality control of the nanopore sequencing technology, which is already known to us. I would expect the authors apply Nanopore sequencing to raise scientific questions and find some new and interesting results to the scientific society, for example, novel splicing isoform that contribute to the cell identity.

The concern of the Reviewer is that no scientific questions are risen in the manuscript and no interesting results are present such as, for example, novel splicing isoforms which could contribute to the cell identity. We kindly disagree with the Reviewer. Perhaps, the question we raised in the manuscript is not biologically straightforward such as could be that related to the issue of finding novel splicing isoforms but it is, in our opinion, still undoubtedly interesting and scientific. The issue of a metric to describe alternative splicing profiles and to present them for further analysis with bioinformatics tools is quite important. The ‘exon usage’ and PSI index are used in short-read RNA-seq because of inherent difficulties in assembling of short reads into transcript isoforms. Surprisingly, these metrics continued to be used in the ONT data analysis (e.g., refs. 23-25 of the revised manuscript) despite the fact that, in the long-read ONT sequencing, the standard treatment of raw sequencing data directly provides the abundance of each and every transcript isoform and, consequently, allows to easily calculate the number of splice variant (transcript isoforms) expressed by each gene. We additionally stressed this point upon revision (lines 242-251). 

Overall, the conception novelty of the manuscript is low. The data presented are just a QC of nanopore sequencing technology without raising scientific questions and deep investigation of the splicing isoforms got from nanopore sequencing. The current form of the manuscript is hard to be published as a research article. I would suggest the authors take extensive investigation of the splicing isoforms and raise more interesting scientific questions rather than just submit their QC results of the sequencing data.

We kindly disagree with the Reviewer that the data presented in the manuscript is just a QC of nanopore sequencing technology. As mentioned above, we suggested in the manuscript to use a novel metric to describe alternative splicing profiles in the case of nanopore sequencing. Since the suggested metric has to allow, as a minimum, for distinguishing between various cells and/or tissue types, based on differences in their AS profiles, we tested the metric by applying PCA to three types of biospecimens expected to differ in alternative splicing, using alternative splicing profiles expressed in the terms of degree of splicing. We did it on the level of whole transcriptome and for a particular set of genes known as pharmacogenes, motivated by the role the liver plays in the drug metabolism.

As response to the Reviewer criticism, we extended our analysis further upon revision to make our manuscript more “biologically sound”. Namely, we dug deeper into PCA data to retrieve genes mostly responsible for the observed clustering of biospecimens in PCA, based on their alternative splicing.  Among these genes we defined subsets of genes whose alternative splicing profiles statistically significantly differ between studied types of biospecimens and subjected them to gene ontology analysis. This analysis revealed that translation is a biological pathway mostly affected by differences in alternative splicing in HepG2, Huh7, and human liver tissue. 

All changes made to the manuscript upon revision are highlighted in yellow.

Reviewer 2 Report

The authors demonstrate a very simple PCA-based method to discriminate  normal tissues and cancer cell lines in terms of 'splicing diversity'. The results are very clear and the message is straightforward, but the authors need to give to the readers more information so to appreciate the biological sense of the work:

1. What is the reason to use 'pharmacogenes' in contrast to global sampling of isoforms ? A corollary of this question is the need of a description of what is intended for pharmacogene. By the way we know that genome expression can be considered as an integrated dynamical system and that any random choice of genes  (when reaching a releatively low threshold of genes) gives more or less the same results (see for example: Sirbu, O., Helmy, M., Giuliani, A., & Selvarajoo, K. (2023). Globally invariant behavior of oncogenes and random genes at population but not at single cell level. npj Systems Biology and Applications9(1), 28., Tsuchyia, Masa, et al. "Gene expression waves: cell cycle independent collective dynamics in cultured cells." The FEBS journal 274.11 (2007): 2878-2886.).

2. Supplementary tables are not visible in Supplementary material but I suppose that they do not include the loading matrix (i.e. the correlation coefficients between original variables and components (see Giuliani, Alessandro. "The application of principal component analysis to drug discovery and biomedical data." Drug discovery today 22.7 (2017): 1069-1076.) loading pattern is of utmost importance to understand the meaning of extracted components. As a matter of fact I suspect that PC1 is a size component (see attached file) with all loadings with the same sign and simply reflects different abundance of isoforms in different specimens (while PC2 is a shape component).

3. This question has to do with the above point: I imagine the authors (given the disproportionate number of isoforms with respect to samples, did use isoforms as statistical units and isoforms as variables and this is suggested by the fact rows are the isoforms, but this point must be clearly underlined even for understanding the measurment unit of plots (are they eigenvectors or scores ?). 

4. Which is a possible explanation of the greater diversity of isoforms in cell lines with respect to normal tissues ? 

If the above issues are taken into consideration and adequately faced I think the manuscript as merits to be published.

Author Response

The authors demonstrate a very simple PCA-based method to discriminate  normal tissues and cancer cell lines in terms of 'splicing diversity'. The results are very clear and the message is straightforward, but the authors need to give to the readers more information so to appreciate the biological sense of the work:

  1. What is the reason to use 'pharmacogenes' in contrast to global sampling of isoforms ? A corollary of this question is the need of a description of what is intended for pharmacogene. By the way we know that genome expression can be considered as an integrated dynamical system and that any random choice of genes (when reaching a releatively low threshold of genes) gives more or less the same results (see for example: Sirbu, O., Helmy, M., Giuliani, A., & Selvarajoo, K. (2023). Globally invariant behavior of oncogenes and random genes at population but not at single cell level. npj Systems Biology and Applications, 9(1), 28., Tsuchyia, Masa, et al. "Gene expression waves: cell cycle independent collective dynamics in cultured cells." The FEBS journal 274.11 (2007): 2878-2886.).

The reason to analyze pharmacogenes was that we used in our study liver tissue and hepatocyte-derived cell lines and that hepatocytes are known to be responsible for the metabolism of the most drugs. The Reviewer is absolutely right that a randomly chosen set of genes can give the similar result. However, this is known for gene expression but not for alternative splicing either in terms of “exon usage”/PSI index or metric suggested in our manuscript. Nonetheless, the Reviewer remark stimulated us to conduct PCA for two randomly chosen subset of genes matching in size those of pharmacogenes. The first subset was randomly selected from the set of gene used in PCA for Figure 2c (whole transcriptome with TPM > 10). The second subset was randomly selected from the set of gene used in PCA for Figure 2f (whole transcriptome with TPM > 10 and no zero DS values for any biospecimen). The result is presented as Figure S-2. It turned out that the first subset of genes did show clustering in PCA similar to that for whole transcriptome. The second subset also demonstrated clustering for all type of biospecimens in contrast to pharmacogenes where such clear clustering was not observed for HepG2 and Huh7 cells (Figures S-? vs. Figure 3f). Thus, when only genes with no zero DS values are involved into analysis (only moderately expressed splice variants are taken into account), the set of pharmacogenes differ from the randomly chosen subset of gene, matching in size. The pertinent discussion is provided upon revision (lines 311-316). We are thankful to the Reviewer for drawing our attention to the relevant papers which have been cited as refs. 21 and 22 upon revision.

  1. Supplementary tables are not visible in Supplementary material but I suppose that they do not include the loading matrix (i.e. the correlation coefficients between original variables and components (see Giuliani, Alessandro. "The application of principal component analysis to drug discovery and biomedical data." Drug discovery today 22.7 (2017): 1069-1076.) loading pattern is of utmost importance to understand the meaning of extracted components. As a matter of fact I suspect that PC1 is a size component (see attached file) with all loadings with the same sign and simply reflects different abundance of isoforms in different specimens (while PC2 is a shape component).

            We apologize for such an inconvenience. The Supplementary material was not visible probably by technical reasons. Indeed, there was no table which would include loading matrix in the original version of manuscript and the supplementary material. Upon revision, we included the table with loadings as Table S-5. The paper which the Reviewer has drawn our attention to was cited as ref. 2 upon revision.

  1. This question has to do with the above point: I imagine the authors (given the disproportionate number of isoforms with respect to samples, did use isoforms as statistical units and isoforms as variables and this is suggested by the fact rows are the isoforms, but this point must be clearly underlined even for understanding the measurment unit of plots (are they eigenvectors or scores ?).

            The Table S-3 which we used as input for PCA is in fact a data matrix with biospecimens as objects (samples) and ENSG identifiers (genes) as variables (descriptors). The values of variables for each object are degree of splicing (the number of splice variant per gene). The results of PCA represented as scores/scores plots (Figures 2 and 3). We made pertinent clarification in legends for Figures 2 and 3 upon revision as recommended by the Reviewer.

  1. Which is a possible explanation of the greater diversity of isoforms in cell lines with respect to normal tissues ?

The cell lines we used in our study are derived from malignant tumors of liver. Thus, on one hand, they can represent a mixture of different clones with not similar profiles of alternative splicing. And this can be a reason for the observed greater diversity of isoforms. On the other hand, the malignant transformation by itself can be associated with increase in alternative splicing events since alternative splicing is known to be one of the key drivers of the formation of cancer phenotype through the production of aberrant splice forms, the formation of splice forms with altered translation rate or direct influence on the diversity of proteoforms. However, in our study, we did not aim at evaluating the reason behind greater diversity of isoforms in malignant cell lines and normal tissue. We intendedly used normal tissue samples and samples of malignant cells originated from different tumors to expect, based on literature data, dissimilar alternative splicing profiles to evaluate the suitability of the suggested metric for studying differences in alternative splicing between various tissues/cells at the level of whole transcriptome or particular subsets of genes.

All changes made to the manuscript upon revision are highlighted in yellow.

Reviewer 3 Report

The authors presents a manuscript focusing on PCA analysis on AS from long-read RNA datasets. Long-read RNA for AS profiling is of great interest. However, the simplified analysis by PCA without deeper understanding of biological relevance and further validation is hard to persuade readers on the analysis's application and paper's value.

Thus, it is highly recommender that the authors dig deeper on the AS profiles besides PCA analysis, to add experiments to validate on what has been proposed from the PCA and further analysis.

acceptable

Author Response

The authors presents a manuscript focusing on PCA analysis on AS from long-read RNA datasets. Long-read RNA for AS profiling is of great interest. However, the simplified analysis by PCA without deeper understanding of biological relevance and further validation is hard to persuade readers on the analysis's application and paper's value.

Thus, it is highly recommender that the authors dig deeper on the AS profiles besides PCA analysis, to add experiments to validate on what has been proposed from the PCA and further analysis.

Following the Reviewer recommendation, we extended our analysis based on results of conducted PCA to dig deeper on the AS profiles. We retrieved genes mostly responsible for the observed clustering of biospecimens in PCA from PSA data. Among these genes we defined subsets of genes whose alternative splicing profiles statistically significantly differed between studied types of biospecimens and subjected them to gene ontology analysis. This analysis revealed that translation is a biological pathway mostly affected by differences in alternative splicing in HepG2, Huh7, and human liver tissue. Consequently, the new subsections (2.3. Subsets of genes determining differences in alternative splicing profiles for liver tissue and hepatocyte-derived cultured cells and 4.4. PCA loadings, ANOVA test, and gene ontology analysis) were added to the manuscripts, alongside with new figures (Figs. 4, S-2, and S-3) and tables (Tables S-5 – S-7). The relevant discussion of new results was also provided upon revision (lines 322-343).    

All changes made to the manuscript upon revision are highlighted in yellow.

Round 2

Reviewer 1 Report

The authors did improved their manuscript by including additional analysis. However, the revised manuscript still focus on the bioinformatic analysis, of which the majority is PCA. The authors seem have difficulty in well elaborating their sequencing data and providing further experimental validation to present an appealing story.

Author Response

The submitted manuscript does focus on the bioinformatic analysis of transcriptome profiles by PCA. However, we kindly disagree with the Reviewer that such analysis does not present an appealing story. The sequencing data can be elaborated in different ways. Aside of focusing on particular transcript isoforms, including finding novel ones, the sequencing data can be converted into transcriptome-wide gene expression profiles. The gene expression profiles are commonly used as a kind of ‘molecular signature’ to differentiate various tissues or cells, or normal and pathologic states. Alongside with expression profiles presented in terms of transcript abundances, the alternative splicing profiles are also used for these purposes. Since presently the overwhelming majority of sequencing data is short-read RNA-seq data, alternative splicing profiles are usually described using metrics such as ‘exon usage’ or PSI index which are the substitutes. To obtain the actual splicing pattern when each gene can be characterized by the number of expressed splice variants (transcript isoforms) and their abundance is challenging task for the short-read RNA sequencing. In contrast, for the long-read RNA sequencing such the ONT nanopore sequencing, the splicing pattern in terms of splice isoform abundance is an intrinsic output of raw data analysis and can be easily and unambiguously converted into splicing patters expressed in terms of the number of expressed splice variants (transcript isoforms) per each and every gene. In our work, we referred to the latter as ‘degree of splicing’ and suggested as a simple metric to describe alternative splicing profiles in the case of nanopore sequencing instead of commonly used ‘exon usage’ or PSI index.

We demonstrated that the suggested metric does work. The metric allowed to distribute biospecimens of liver tissue and hepatocyte-derived malignant cells into clear clusters in PCA. When we retrieved genes mostly responsible for the observed clustering of biospecimens in PCA from PCA data. Among these genes we defined subsets of genes whose alternative splicing profiles statistically significantly differed between studied types of biospecimens and subjected them to gene ontology analysis. This analysis revealed that translation is a biological pathway mostly affected by differences in alternative splicing in HepG2, Huh7, and human liver tissue.

As response to the Reviewer criticism, we focused on transcript isoforms expressed by the selected genes. We additionally find among them isoforms which are expressed only in one type of biospecimens (at a given TPM threshold) and trace whether they are translated into proteins. The 12 transcripts isoforms out of 18 found coded either canonical or non-canonical protein isoforms which do expressed and can potentially influence the particular biological pathway and, consequently, contribute to the observed phenotypes of studied biospecimens.

Consequently, the new subsection (2.4. Transcripts isoforms involved in translation and characteristic for liver tissue, HepG2 or Huh7 cells) were added to the manuscripts, alongside with new figures (Figs. S-4 and S-5). The relevant discussion of new results was also provided upon revision (lines 369-381).

All changes made to the manuscript upon revision are highlighted in yellow.

Reviewer 2 Report

The authors did take into considerations the issues I raised and I think that the revised manuscript deserves publication

Author Response

We are thankful to the Reviewer for positive evaluation of our work and for critical comments and suggestions which stimulated us to improve the manuscript.

Reviewer 3 Report

thanks for the responses

Author Response

(The authors gave the same response as above.)

Round 3

Reviewer 1 Report

Thanks for the author's explanation.